# Assessing the Capacity of Adaptive Policy Pathways to Adapt on Time by Mapping Trigger Values to Their Outcomes

**Luciano Raso** \*, **Jan Kwakkel** and **Jos Timmermans**

Department of Multi-Actor Systems, Delft University of Technology, Jaffalaan 5, 2628 BX Delft, The Netherlands; J.H.Kwakkel@tudelft.nl (J.K.); J.S.Timmermans@tudelft.nl (J.T.)
\* Correspondence: luciofaso@gmail.com

**Abstract:** Climate change raises serious concerns for policymakers that want to ensure the success of long-term policies. To guarantee satisfactory decisions in the face of deep uncertainties, adaptive policy pathways might be used. Adaptive policy pathways are designed to take actions according to how the future will actually unfold. In adaptive pathways, a monitoring system collects the evidence required for activating the next adaptive action. This monitoring system is made of signposts and triggers. Signposts are indicators that track the performance of the pathway. When signposts reach pre-specified trigger values, the next action on the pathway is implemented. The effectiveness of the monitoring system is pivotal to the success of adaptive policy pathways, therefore the decision-makers would like to have sufficient confidence about the future capacity to adapt on time. "On time" means activating the next action on a pathway neither so early that it incurs unnecessary costs, nor so late that it incurs avoidable damages. In this paper, we show how mapping the relations between triggers and the probability of misclassification errors inform the level of confidence that a monitoring system for adaptive policy pathways can provide. Specifically, we present the "trigger-probability" mapping and the "trigger-consequences" mappings. The former mapping displays the interplay between trigger values for a given signpost and the level of confidence regarding whether change occurs and adaptation is needed. The latter mapping displays the interplay between trigger values for a given signpost and the consequences of misclassification errors for both adapting the policy or not. In a case study, we illustrate how these mappings can be used to test the effectiveness of a monitoring system, and how they can be integrated into the process of designing an adaptive policy.

**Keywords:** climate change; adaptation; monitoring; flood protection; Afsluitdijk; extremes; changing extremes; adaptive policies; resilience; risk management

## 1. Introduction

Despite all research efforts, climate change remains unpredictable in the long term, raising serious concerns for policymakers that want to ensure the success of long-term policies [1]. Adaptive policy pathways [2] have recently emerged as a way to increase the effectiveness of long-term policies in the face of the unavoidable "deep uncertainties" [3]. Adaptive policy pathways is one method for developing adaptive policies; other methods include Assumption-Based Planning [4], Dynamic Adaptive Policies [5], Real Options [6–9], Adaptive Policy-Making [10–12], Adaptation Options [13], Adaptation Tipping Points [14], and Adaptation Pathways [15–17].

In designing adaptative pathways, the analyst explores the consequences of multiple scenarios, often by use of a system model [18,19]: these scenarios represent the multiple possible future evolutions of the system. Then, the analyst can assemble a long-term plan of action that can respond to a large

set of possible future scenarios. This plan of action is made of multiple sequences of concatenated actions: a new action is activated if its predecessor action is no longer able to guarantee policy success. Advantages of adaptive policies are their capacity to value correctable (or scalable) decisions, modulate response to evidence of change, coordinate short- and long-term actions, and delay decisions to keep future options open [20–22].

The effective implementation of an adaptive pathway over time depends critically on a monitoring system. This monitoring system tracks the information required for the timely activation of the next action on the pathway. The ability of the monitoring system to effectively detect change is pivotal to the success of the whole pathway. The information required to establish the need for adaptation, however, can be noisy and ambiguous [23], inhibiting the capacity to activate the next action on time [22]. For example, distinguishing change in flood risk from the natural variability is hampered by the scarcity of valuable data-points. A well-designed monitoring system should gather enough evidence for confident adaptation on time. "On time" means adapting the policy neither so early that it incurs unnecessary costs, nor so late that it incurs avoidable damages. Despite its importance, the effectiveness of this monitoring system is rarely tested before the policy is implemented.

The literature suggests basing the monitoring system of adaptive policies on "signposts" [4] and "triggers" [12]. Signposts are indicators that identify the information that should be tracked to determine whether the policy is meeting the conditions for its success, and triggers are specific values of signposts that identify critical values beyond which additional pre-specified actions should be implemented [2,5,12]. An example of a signpost, later used in the test case, is the average sea level rise rate at a certain location over a window of 10 years. An example of trigger for this signpost is 10 mm/year.

Refs. [23,24] identify some criteria to assess the quality of a monitoring system and how to employ a system model in their evaluation, and [25] tested the detectability over time of a change in the 100-year flood. Despite these initial research efforts, the literature on monitoring systems for adaptive policies is still under-developed. In particular, we see a lack of methods and instruments to evaluate whether a monitoring system is sufficiently effective in collecting the evidence of change "on time", such that the appropriate adaptive action can be taken with a sufficient level of confidence.

In this paper, we show how mapping trigger values against their outcomes is a way to assess the expected confidence about the capacity to adapt the policy on time, and its consequences. We introduce two mappings, (i) the *trigger-probability* mapping and (ii) the *trigger-consequences* mapping. The trigger-probability mapping displays the interplay between trigger values for a given signpost and the level of confidence regarding both that critical conditions have changed, and that adaptation is needed. The trigger-consequences mapping displays the interplay between trigger values for a given signpost and the consequences of misclassification for both adapting or non-adapting the policy. We illustrate how these mappings can be used to test the effectiveness of a monitoring system, and how they can be integrated into the process of designing an adaptive policy.

The paper is structured accordingly: in the Methodology we first describe the problem of monitoring in adaptive policies then we explain how to identify and how to interpret the proposed mappings; in the Application we demonstrate the use of the instruments on a test case of an adaptive policy for costal flood protection in the Netherlands; we then present our summary and discussion in the Conclusions.

## 2. Methodology

The monitoring system of adaptive policies generally consists of signposts and triggers. Signposts are indicators designed to track the exogenous developments that affect the performance of a plan. When a signpost exceeds its trigger value, it activates an adaptation signal. Adapting on time requires activating the adaptation signal when the system approaches an adaptation tipping point, but not too early. An adaptation tipping point represents the conditions under which a policy no longer meets its specified objectives, hence additional or distinct actions are needed [14]. An example of

adaptation tipping point, later used in the application, is when the flood risk exceeds a pre-specified acceptable value. A good monitoring system should be able to provide a timely adaptation signal, i.e., to be activated in proximity of an adaptation tipping point. Testing the capacity provides a timely adaptation signal requires estimating the level of confidence in the adaptation signal at the adaptation tipping point.

Signpost estimates are observations intended to detect the real state of the system. Signpost estimates, however, can be noisy, and so will be the adaptation signal, resulting in possible misclassification errors. Table 1 presents all possible consequences for the combination of the real states of the system (columns) and the adaptation signal (rows).

**Table 1.** Table of Consequences (*Probability*).

| Consequence (*Probability*) | | Adaptation Required | |
|---|---|---|---|
| | | **Yes** | **No** |
| Adaptation Signal | Yes | Timely Adaptation (*Power*) | Regretful Action (*Significance*) |
| | No | Missed Adaptation (*1-Power*) | No Adaptation (*1-Significance*) |

Adaptation is required when the state of the system is at the adaptation tipping point. The adaptation signal is activated when the signpost exceeds the trigger value. "Timely Adaptation" and "No Adaptation" are the correct decisions. "Missed Adaptation" and "Regretful Action" are misclassification errors stemming from the uncertainty of the signpost estimate. Missed adaptation and regretful action correspond to Type I and Type II errors in hypothesis testing [26].

Type I and Type II errors are related to "significance" and "power" of a statistical test. Significance is the probability of Type I error. In the context of climate adaptation, significance is the probability of erroneously activating the adaptation signal when the critical uncertainties did not change. Critical uncertainties are conditions that are presently uncertain or subject to change in the future, which strongly affect the success of the policy. An example of critical uncertainty, later used in the application, is the future sea level rise rate. Power is the complementary of the probability of Type II error. Power is the probability of correctly detecting the need for adaptation when this is required, i.e., the critical uncertainties are at an adaptation tipping point. In climate change adaptation, both types of errors may have severe consequences, therefore both power and significance are to be kept under control, i.e., as close as possible to one and to zero.

Equation (1) shows how to estimate significance and power for a signpost at a specific adaptation point.

$$\text{Significance} = P(S \geq S_T | \lambda_{CC}) \tag{1a}$$

$$\text{Power} = P(S \geq S_T | \lambda_{ATP}) \tag{1b}$$

In Equation (1), $S$ is the signpost estimate, $S_T$ is the trigger value, $\lambda_{CC}$ the critical uncertainties at the current conditions, and $\lambda_{ATP}$ the same critical uncertainties at the considered adaptation tipping point. Significance is the probability that the signpost estimate extracted from $P(S|\lambda_{CC})$ exceeds a given trigger value. Power is the probability that the signpost estimate extracted from $P(S|\lambda_{ATP})$ exceeds a given trigger value.

$P(S|\lambda_{CC})$ is the observational uncertainty of the signpost at current conditions, and and $P(S|\lambda_{ATP})$ is the observational uncertainty of the signpost at the adaptation tipping point. The signpost observational uncertainty at current conditions can be estimated from observed data by identifying a stochastic model. In the simplest case, the stochastic model is a probability distribution. Estimating the signpost observational uncertainty at the adaptation tipping point, instead, requires the combined use of exploratory and stochastic modeling approaches: exploratory modeling [27,28] is used to represent

the system behavior at the adaptation tipping point of interest, while stochastic modeling [29,30] is used to represent the uncertainty at that point.

In monitoring of adaptive policies, significance and power depend both on the quality of a signpost and on the selected trigger value. A good signpost improves both significance and power. Trigger value selection, instead, is a trade-off between the risks of regretful adaptation and missed adaptation. Selection of signposts is closer to a purely technical choice, being about maximizing the extraction of information from data [23]. Selection of trigger values is partially a political issue, for the presence of multiple actors implies a redistribution of benefits and costs related to the adaptation decision, hence conflicting interests about the level of confidence required for implementing the adaptive actions. Having identified Equation (1) opens the possibility of exploring the interplay between trigger values and their outcomes.

The trigger-probability mapping explores how trigger value maps to significance and to the complement of power. The trigger-probability mapping is identified by estimating significance and power, as in Equation (1), for all possible values of the trigger. Figure 1 visually displays an example of trigger-probability mapping based on an idealized case. In Figure 1 it is highlighted how significance corresponds to the probability of Type I error and power corresponds to the complement of probability of Type II error. Figure 1 shows how significance decreases, and the inverse of power increases with increasing trigger value.

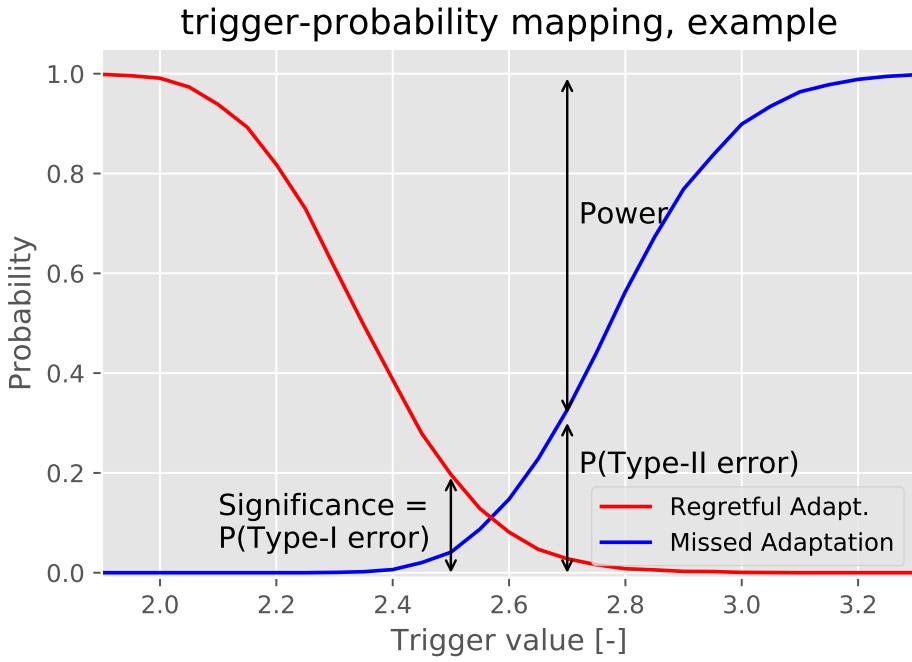

**Figure 1.** Visual representation of trigger-probability mapping, example.

The trigger-probability mapping offers the relevant information to evaluate the current monitoring system and its capacity to offer a timely adaptation, giving indication about (i) the signpost quality, and (ii) the trigger values candidate to the final selection. The trigger-probability mapping shows how, when the trigger value increases, the probability of Type I error (i.e., significance) decreases, and the probability of Type II error (i.e., the complement of power) increases. If this occurs simultaneously, i.e., for the same value of the trigger, it is not possible to keep both classification errors low, for lowering significance happens at the cost of lower power. In this case, the signpost has low quality, i.e., the signpost does not provide the information required with sufficient confidence. If, as in Figure 1, there is a space of trigger vales in which both probability of Type I and Type II errors are considered sufficiently small, then the signpost is capable of detecting the required adaptation at the

considered adaptation tipping point with sufficient confidence, and an appropriate trigger value will likely be in this space.

When the consequences of adaptation decisions are available, as in Table 1, the *trigger-consequences* mapping can be built. The trigger-consequence mapping explores how trigger value maps to the expected consequences of misclassification errors. The consequences of misclassification errors are (i) the consequences for regretful adaptation weighted by the probability of Type I error, and (ii) the consequences for missed adaptation weighted by the probability of Type II error. As for the trigger-probability mapping, the trigger-consequences mapping is specific to a signpost and a selected tipping point. The trigger-consequence mapping offers information about the expected consequences of adapting or not, and it can be used to estimate the potential value of adaptive policies. Different groups of stakeholders may be interested in different types of consequence; therefore, mapping trigger values to consequences for each group of stakeholders, and all possible trade-offs, can support the process of selecting the final trigger value.

Figure 2 visually displays an example of trigger-consequence mapping, where the cost of the adaptation action is 1 M$, and the consequence of non-adaptation is 2 M$. The lowest value of trigger corresponds to a case when the adaptation action is always taken. In this case, there is no risk of missed adaptation, but high possible regret, equivalent to the full cost of the adaptation action, if the action turns out to be non-necessary. This situation can be considered as equivalent to using static robust decision-making policies [31]. The highest trigger value corresponds to a case where the action for adaptation is never taken. In this case, costs are related to consequences of missed adaptation. If a space of trigger values exists for which one can limit both consequences of regretful action and missed adaptation, then adaptive policies can offer a potential benefit.

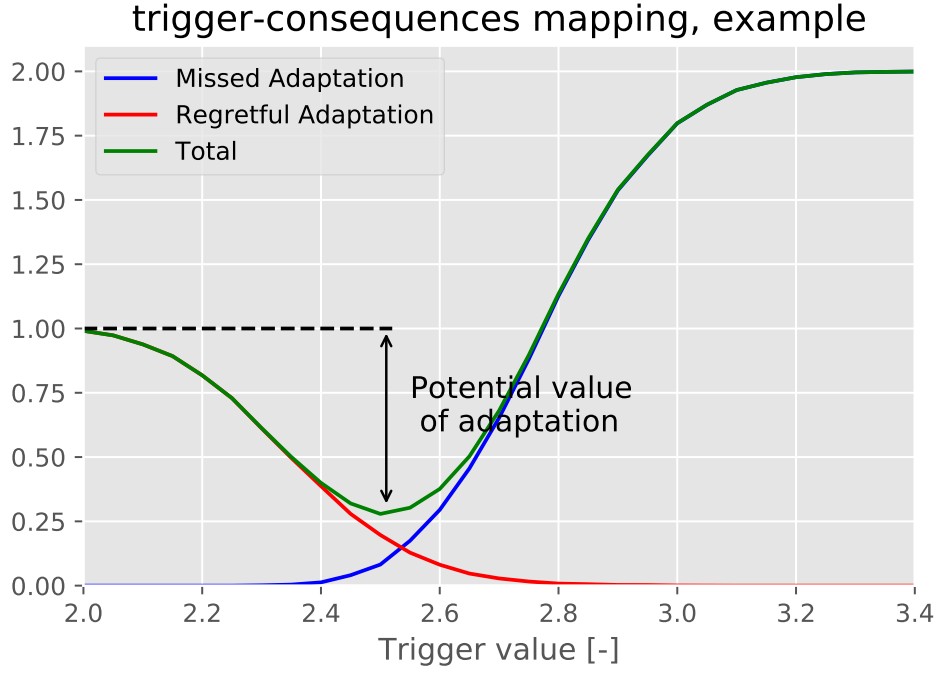

**Figure 2.** Visual representation of trigger-consequences mapping, example.

Figure 2 presents a case where the consequences of regretful action (red line) and missed adaptation (blue line) can be aggregated in total costs (green line); for example, because both are expressed in monetary terms under the assumption of no risk aversion. In this case, the difference between the consequences for regretful action and the lowest point in the total costs can be considered to be the expected potential value of adaptation (see Figure 2). If there is no space in which both expected consequences of regretful and missed adaptation can be kept low, then the present monitoring system does not offer sufficient evidence to detect the adaptation tipping point on time. In this situation,

the risks of adaptation outweigh the benefit of uncertainty reduction through an adaptive policy, and it may be preferable to resort to static robust decision-making.

The trigger-probability and the trigger-consequences mapping can be used to assess whether an adaptive policy and its monitoring system offer sufficient confidence about the future capacity to adapt on time for all considered future scenarios. If this is not the case, the adaptive policy must be reconsidered, possibly by improving the monitoring system, or by redefining the plan of actions. Improving the monitoring system requires the identification of more informative or robust signposts. This may be not always feasible; for example, all available information may already have been exploited, or  the costs of gathering additional information may outweigh its benefits [32]. In this case, effective adaptation cannot be guaranteed, and it may be necessary to redesign the plan of action such that static robust actions are included.

## 3. Application

The use of the proposed methodology is demonstrated on an application of adaptive policy for costal flood protection in the Netherlands. In the following, we present the policy problem, the critical uncertainties, the adaptive policy with its related monitoring system, and its evaluation by use of the proposed instruments. The model and the details of calculation to derive the trigger-probability and trigger-consequences mappings are presented in the Appendix A–C, including the reference to the code (Appendix D), which is available and reusable.

### 3.1. Policy Problem

In the Netherlands, the most inhabited and economically valuable areas are also the most vulnerable to flood risk.  A complex system of dikes protects the Randstad, the economic core of the country, which lays below sea level.  In this application we focus on Afsluitdijk, a 32 km dam that functions as storm surge barrier, separating the Wadden Sea from Lake Ijssel. The national flood safety standards for Afsluitdijk require a level of protection of 1 in 10,000 years, Recently, a national assessment program concluded that Afsluitdijk does not offer the required level of protection, therefore Afsluitdijk will be renovated in the near future.

### 3.2. Critical Uncertainties

In this application we focus on critical uncertainties related to climatic conditions. The climatic uncertainties that are critical for Afsluitdijk are related to the hydraulic boundary conditions that may change because of climate non-stationarity, and on which the level of protection of the dike depends. Specifically, we consider the following critical uncertainties:

- Rate of sea level rise
- Storm surge intensity

The presence of a clear trend in sea level rise is well documented [33]. The rate at which the sea level rise will take place, however, is deeply uncertain [34]. In the next century the rate of the sea level rise can increase or, if global policies to reduce the effect of climate change are effective, remain stable [33].

Frequency and intensity of extreme storm surge events influence the level of protection offered by Afsluitdijk. The literature on climatic change detection indicates that storm surge intensity may change as a consequence of climate change, as well as wave height characteristics and spectral wave period, and that change is particularly marked for extremes events [35–37]. The literature, however, analyzes the climatic change at the global level, or relative to other regions [38–44], whereas we are interested in the change in storm surge intensity around Afsluitdijk.

Climate change predictions show no clear trend in increase storm surge intensity [45] in the region of interest. Such "top-down" analysis, however, derives its results from climate change projections by use of a chain of models, whose skills in correctly representing extreme events are questionable,

therefore results from this type of analysis cannot be used to exclude storm surge intensity from the set of critical uncertainties regarding the future performance of Afsluitdijk.

### 3.3. Adaptive Policy

We consider an adaptive dike design created from features of both the "2010 Robust Alternative", and the "Wadden Werken" alternative [46]. Figure 3 shows the pathways map of the adaptive policy and a schematic section of the dike and the alternative action for each pathway.

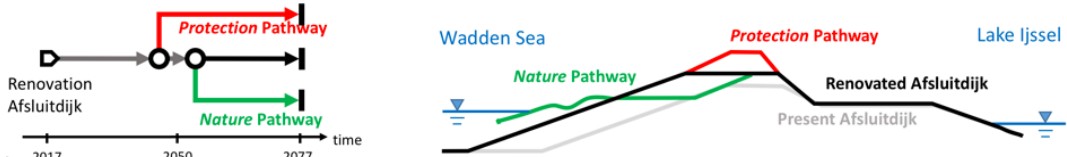

**Figure 3.** (**Left**) Adaptive pathways map for the renovated Afsluitdijk, including Protection Pathway (red), Zero Pathway (black), and Nature Pathway (green). (**Right**) schema of dike section showing alternative pathways.

We consider three possible adaptation pathways. In "Zero Pathway", Afsluitdijk retains its renovated form. In "Protection Pathway", a dike reinforcement is implemented by heightening the dike crown. In "Nature Pathway", the mass of the dike is reduced by removing part of the outer dike core, adding a berm on the external side, and implementing building-with-nature solutions that increase the natural and recreational value of the dike [47]. The Protection or Nature Pathways are followed when the risk of dike failure turns out to be larger or smaller than initially expected. The adaptation decision is expected to be taken around 2050. This is in line with the current policy recommendation [48]. The adaptation tipping point related to risk of dike failure are set to 1 in 5000 years for the Protection Pathway, and 1 in 12,000 years for the Nature Pathway.

Figure 4a presents the "stress test" [49] for the renovated Afsluitdijk for a range of possible values of critical uncertainties, evaluated in 2050. This figure shows the area of success for the renovated Afsluitdijk, delimited by the boundaries in which the expected frequency of failure is at the adaptation tipping points. Results of Figure 3 are obtained from the van-der-Meer model [50] used in an exploratory analysis, i.e., tested on a large set of possible critical uncertainties. The van-der-Meer model and the parameters for Afsluitdijk are fully described in the Appendix A.

### 3.4. Design of the Monitoring System

Table 2 describes the candidate signposts and the parameters that each signpost is intended to track. Data will be obtained from the station at den Oever, which is the sea level gauging station closest toAfsluitdijk.

**Table 2.** Initial candidate signposts.

| Signpost | Definition | Parameter |
|:---:|---|:---:|
| $S_A$ | Sea level rise rate at den Oever, on a 10-year moving window average | Average sea level rise rate |
| $S_E$ | Yearly maximum water level at den Oever, on a 20-year moving window average | Yearly maximum water level |

The identification of the trigger-probability and the trigger-consequences mapping requires selection of the adaptation tipping points and identifying the signpost uncertainty at both the adaptation tipping points and at current conditions. The following four adaptation tipping points are selected.

$\lambda_{SLR+}$    Increase in average sea level rise rate, constant storm intensity.

$\lambda_{SS+}$　　Increase in storm surge intensity, constant average sea level rise rate.

$\lambda_{SLR-}$　　Reduction in average sea level rise rate, constant storm.

$\lambda_{SS-}$　　Reduction in storm surge intensity, constant average sea level rise rate.

All adaptation tipping points are on the boundary of the policy success region, as represented in the stress test on Figure 4a. Tipping points $\lambda_{SLR+}$ and $\lambda_{SS+}$ represent an increased flood risk. Reaching any linear combination of these tipping points activates the Protection Pathway. Tipping points $\lambda_{SLR-}$ and $\lambda_{SS-}$ represent a reduced flood risk. Reaching any linear combination of these tipping points activates the Nature Pathway. Estimation of $P(S|\lambda)$ at these tipping points and at current conditions is described in the Appendix A.

The selected tipping points explore the change of one critical uncertainty at a time. Change, however, is likely to occur simultaneously on both critical uncertainties. We consider these two couples of points as extreme cases of the tipping point surface: if the monitoring system can detect change for these two limit conditions, it is likely that it can detect change also for a combination of them.

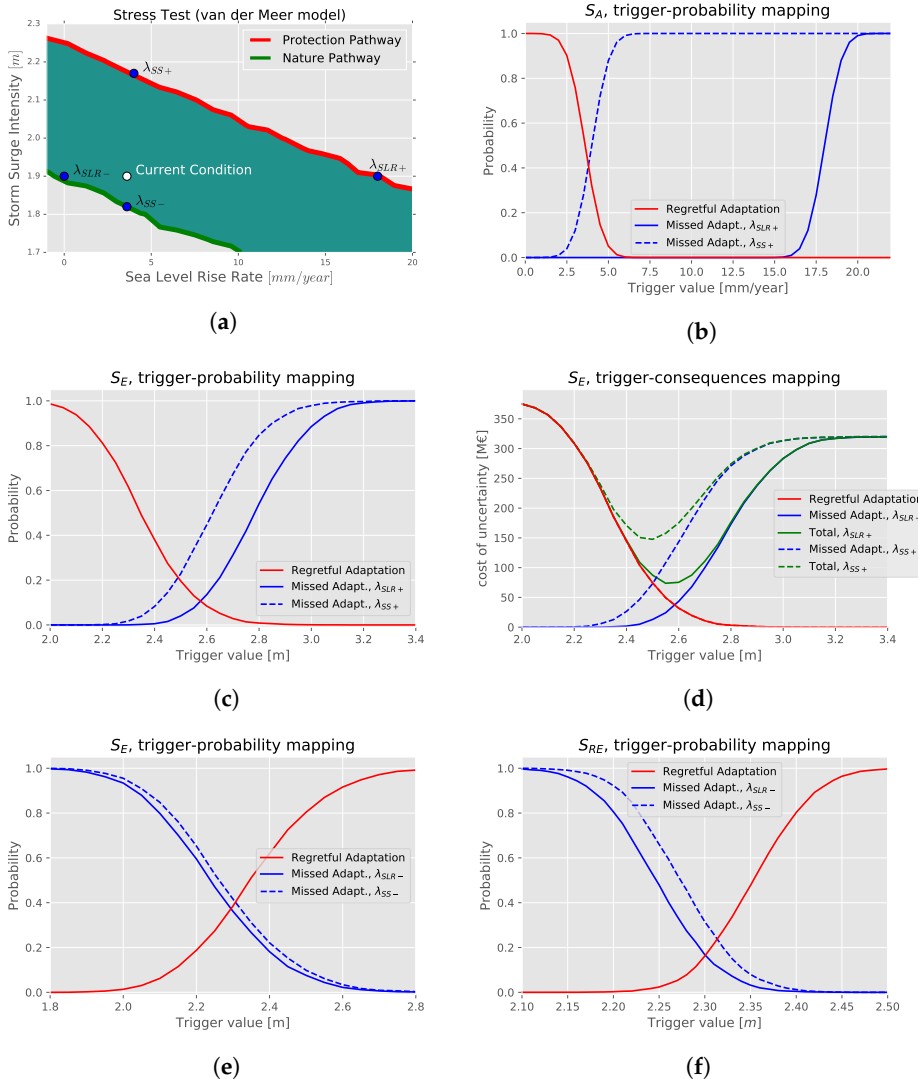

**Figure 4.** (**a**): Stress-test of the renovated Afsluitdijk in 2050. Plots (**b**) and (**c**): trigger-probability mapping for signposts $S_A$ and $S_E$ evaluated at the "Protection Pathway" adaptation tipping points. Plot (**d**): trigger-consequences mapping for signpost $S_E$ evaluated at the "Protection Pathway" adaptation tipping points. Plot (**e,f**): trigger-probability mapping for signposts $S_E$ and $S_{RE}$ evaluated at the "Nature Pathway" adaptation tipping points.

## 4. Results

Figure 4b shows the trigger-probability mapping for signpost $S_A$, evaluated at $\lambda_{SLR+}$ and $\lambda_{SS+}$. Figure 4b presents a large plateau between the risk of regretful adaptation and that of missed adaptation for $\lambda_{SS+}$ (blue dashed line), and a complementary trend for $\lambda_{SLR+}$ (blue continuous line). Figure 4b suggests that the signpost $S_A$ can detect $\lambda_{SLR+}$ with very little uncertainty, but it is not able to detect if the system reaches $\lambda_{SS+}$: the adaptation tipping point is almost indistinguishable from the current conditions. The interpretation of Figure 4b implies that $S_A$ is an informative signpost, under the assumption that the rate of sea level rise is the only critical uncertainty that may change. $S_A$, however, is not a robust signpost, because if storm intensity changes, this signpost does not longer offer satisfactory results.

Figure 4c shows the trigger-probability mapping for signpost $S_E$, evaluated at $\lambda_{SLR+}$ and $\lambda_{SS+}$. Figure 4c presents a range of trigger values for which the probability of both misclassification errors is low, which we consider sufficient. This is valid for both the considered adaptation tipping points. Figure 4c suggests that $S_E$ is informative in detecting the need for adaptation and robust to change of both critical uncertainties.

Figure 4d shows the trigger-consequence mapping for signpost $S_E$, evaluated at $\lambda_{SLR+}$ and $\lambda_{SS+}$. The estimation of consequences for missed adaptation and regretful action are described in additional online material, and it is limited to the economic costs for which we have data [51]. Figure 4d indicates the presence of a space of trigger values in which both the cost of regretful adaptation (red line) and the cost of missed adaptation (blue lines) can be kept low. This space is larger for the adaptation tipping point related to the change in sea level rise rate, $\lambda_{SLR+}$ (continuous lines), because this change is easier to detect. Figure 4d shows the total costs (green lines), assuming that costs for regretful action and costs for missed adaptation can be summed up. In this case, a valley and a minimum value can be detected for both adaptation tipping points. The minimum total cost occurs for trigger values ranging between 2.3 and 2.6 m, depending on which adaptation tipping point we are referring to. This range indicates the space of appropriate trigger values.

Figure 4e shows the trigger-probability mapping for signpost $S_E$, evaluated at $\lambda_{SLR-}$ and $\lambda_{SS-}$, i.e., the adaptation tipping points corresponding to the Nature Pathway. Tipping points $\lambda_{SLR-}$ and $\lambda_{SS-}$ are relative to a level of protection of 1 in 12,000-year. In Plot 4e, the line representing the probability of missed adaptation is on the left of the line representing significance because the adaptation signal is activated when the signpost estimate is smaller than the trigger. Figure 4e shows that power and significance change almost simultaneously: there are no values of trigger that can keep both error types low, hence a smaller significance can be obtained only at the cost of smaller power.

The analysis indicates that signpost $S_E$ is effective in detecting the tipping points relative to the level of protection of 1 in 5000 years required to adapt to an increased flood risk, but it is not able to detect the tipping points relative to the level of protection of 1 in 12,000 years with a level of confidence that we consider sufficient. In this case, either the monitoring system or the plan of actions should be redesigned. Following up on this conclusion, we attempt to improve the monitoring system by including an additional signpost.

The additional signpost $S_{RE}$, defined as the "Corrected yearly maximum water level at den Oever and 4 neighboring stations, on a 20-year moving window average", is similar to $S_E$, but it is based on a regional analysis [52]. In the signpost definition, "corrected" refers to the fact that data are transformed to fit to the local conditions at Afsluitdijk (i.e., the closest gauging station, at den Oever). We do not consider the possible correlation among stations: the presence of correlation would reduce the level of confidence that can be obtained from the same data, requiring more stations to reach the same level of observability as compared to the case without correlation.

Figure 4f shows the trigger-probability mapping for signpost $S_{RE}$, evaluated at $\lambda_{SLR-}$ and $\lambda_{SS-}$. Figure 4f indicates the presence of a range of trigger values for which significance can be kept small and power can be kept large. This is valid for both adaptation tipping points. Nonetheless, a non-negligible risk of misclassification error is still present, even when the trigger values are properly selected.

The added value of signpost $S_{RE}$, however, can be better appreciated when Plot Figure 4f is compared to the trigger-probability mapping of signpost $S_E$ at the same adaptation tipping points, displayed in Figure 4e. In Figure 4f the curves of missed adaptation and regretful adaptation are steeper, resulting in a higher capacity for this signpost to detect the need for adaptation.

## 5. Discussion and Conclusions

In this paper, we presented how mapping trigger values to their outcomes is a way to assess the effectiveness of a monitoring system for adaptive policy pathways. Specifically, we introduced the trigger-probability mapping and the trigger-consequences mappings. The trigger-probability mapping displays the interplay between trigger values for a given signpost and the level of confidence regarding both (i) that critical conditions have changed (i.e., the trigger-significance mapping) and (ii) that adaptation is needed (i.e., the trigger-power mapping). The trigger-consequences mapping displays the interplay between trigger values for a given signpost and the consequences of misclassification for both (i) adapting the policy (i.e., the trigger-regretful adaptation mapping) or (ii) non-adapting the policy (i.e., the trigger-missed adaptation mapping). These mappings can be used to analyze the level of confidence that a monitoring system, made of signposts and triggers, can offer about the need for adaptation. Given this information, the decision maker can decide whether such level of confidence is acceptable or not.

We demonstrated the use of both mappings for the ex-ante evaluation of a monitoring system of an adaptive policy pathway for the renovated Afsluitdijk, a storm surge barrier in the Netherlands. In this application, different signposts are tested on their capacity to detect whether critical uncertainties reached an adaptation tipping point. This analysis provides information on signpost effectiveness. The mappings also indicate candidate trigger values valid for the final selection.

The current literature on climate change adaptation overlooks how limitations of the monitoring system reduce the effectiveness of adaptive pathways. The proposed mappings can be used to test whether a candidate monitoring system effectively tracks the required information with sufficient confidence for timely adaptation. We show how to derivate the trigger-outcome mappings from a system model and a given adaptation tipping point, and how to integrate the mappings in the general design procedure of adaptive pathways. The proposed mappings are intended to be applied at the design phase of an adaptive pathway, to offer sufficient guarantees about the future capacity of the pathway to be adapted on time.

The trigger-outcome mapping is obtained from the estimation of significance and power, i.e., the probability of misclassification error of Type I and the complementary of probability of misclassification error of Type II. Studies on climate change detection often use significance as key parameter to investigate and communicate the level of confidence about the modification of climatic conditions that emerges from data. The use of significance is justified by considering that the burden of proof of a specific statement lies upon who makes that statement (in this case, the modification of existing climatic condition). In climate change adaptation policies, however, the costs of non-action are generally as large as the costs of regretful actions. In this case, the use of power, as in the proposed method, can offer relevant information for decision-making. The consideration of both significance and power, and their consequences, enhances the possibility of finding an appropriate trade-off between costs related to misclassification errors. Estimating power, however, requires a system model able to represent the uncertainty about the signpost estimate, to be used in an exploratory mode.

In coastal flood risk, most of the concerns for climatic change are related to sea level rise, and the possible change in its rate. Despite the relevance of sea level rise rate, other relevant parameters may be affected by climate change, and particularly under extreme conditions, such as the change in storm surge intensity level. Change of extreme conditions, however, is more difficult to detect, constituting a limitation for timely adaptation. In the application of the proposed methodology to the renovated Afsluitdijk we include the possible change in storm surge intensity, and we show how the selection of appropriate signposts could enhance the capacity to detect this change.

In the application of the mappings, a reduced number of critical uncertainties are considered. Despite this simplification, the test case is non-trivial. The mappings perform satisfactorily, but their effectiveness in more complex cases is still to be tested. In this application, the mappings seem to maintain their effectiveness when complexity increases. Future research will aim at offering more insight on the validity of the proposed mappings when applied to more complex cases.

The mappings are specific to a signpost and an adaptation tipping point. When the success of an adaptive policy hinges on multiple critical uncertainties, there will be multiple adaptation tipping points, hence the boundary between policy success and policy failure is a surface, rather than a single point. The selection of the appropriate adaptation tipping points from this surface becomes a subjective choice. This limitation originates, however, from the limits of adaptation tipping points in meaningfully representing the real boundaries of the policy success. Nonetheless, the exploration of the adaptation tipping surface can be made by testing multiple points, which can be represented in the plots by multiple lines. In this case, the value of power (i.e., the curve of missed adaptation) is to be identified for multiple adaptation tipping points.

The mappings are built up from the elicitation of a system model, which itself is defined by a set of assumptions, therefore the estimation of the mappings depends on all the model assumptions. In the exploratory modeling approach, assumptions in which we have little confidence of their capacity to hold in the future are treated as critical uncertainties. Nonetheless, the occurrence of a surprising unexpected future, i.e., beyond the space of critical uncertainties, is always possible. The model used in the test case, for example, considers the possible shift in extreme storm surge, i.e., a possible change in the location parameter of the extreme value distribution, but not the possible change in scale and shape parameters.

The analysis should quantify and represent any uncertainty in the mapping estimation. In the application, for example, we represent the multiple possible adaptation tipping point, represented as multiple "missed adaptation" curves. Any additional uncertainty reduces the capacity in detecting the need for adaptation, which results in a reduced potential value of adaptation. Despite any effort to include all uncertainty in the mapping estimation, however, it will be never satisfactory. Even the best-conceived monitoring system is jeopardized by the so-called unknown unknown, i.e., future scenarios for which we are presently not aware of. In this case, a redesign of adaptive policy and the monitoring system itself may be required.

**Author Contributions:** Conceptualization, L.R.; methodology, L.R.; software, L.R.; validation, L.R.; formal analysis, L.R.; investigation, L.R.; resources, L.R.; data curation, L.R..; writing—original draft preparation, L.R.; writing—review and editing, L.R., J.K. and J.T.; visualization, L.R.; supervision, J.K.; project administration, J.K.; funding acquisition, J.T.

**Funding:** This research was funded by the Nederlandse Organisatie voor Wetenschappelijk Onderzoek, i.e., Netherlands Organisation for Scientific Research.

**Conflicts of Interest:** The authors declare no conflict of interest.The funders had no role in the design of the study; in the collection, analyses, or interpretation of data; in the writing of the manuscript, or in the decision to publish the results.

## Variables

| | |
|---|---|
| $SLR$ | Sea level rise rate [m/year] |
| $SS$ | Storm surge intensity [m] |
| $SWL_y$ | Yearly average sea water level (above NAP) in year y [m] |
| $MWL$ | Maximum sea water level (above NAP) [m] |
| $h_s$ | Wave height characteristics [m] |
| $T_m$ | Spectral wave period [s] |
| $q_{occ}$ | Occurring overtopping discharge [m$^3$/m /s ] |
| $q_{crit}$ | Critical overtopping discharge [0.1 m$^3$/m /s] |
| $s_0$ | Wave steepness [-] |
| $RC$ | Crest free board [m] |

| | |
|---|---|
| $CH$ | Dike crown height [m] |
| $\xi$ | Relative wave run-up [m] |
| $g$ | Acceleration constant [9.8 m$^2$/s] |
| $b$ | Berm coefficient [1] |
| $C_1$ | Hydraulic conditions coefficient [2.7 m] |
| $C_2$ | Hydraulic conditions coefficient [1.7 m] |
| $C_3$ | Hydraulic conditions Coefficient [0.1 s$^2$/m] |
| $C_4$ | Hydraulic conditions coefficient [$\mathcal{N}(\mu = 4.7, \sigma = 0.55)$] |
| $C_5$ | Hydraulic conditions coefficient [$\mathcal{N}(\mu = 2.3, \sigma = 0.35)$] |
| $m_o$ | Occurring discharge uncertainty coefficient [1] |
| $m_c$ | Critical discharge uncertainty coefficient [1] |
| $CH$ | Dike crown height (above NAP) [7.85 m] |
| $\mu_{SS}$ | Storm surge location parameter [m] |
| $\sigma_{SS}$ | Storm surge scale parameter [0.394 m] |
| $\beta$ | Angle of wave attack coefficient [1] |
| $f$ | Roughness coefficient [0.9] |
| $q_{crit}$ | Dike overtopping critical discharge [0.1 m$^3$/m/s] |
| $\sigma_{SLR}$ | Standard deviation of yearly average sea water level observations [6 mm] |

## Appendix A. System Model

In the considered adaptive policy for Afsluitdijk, adaptation is required if the probability of dike failure is larger than 1/5000 years or lower than 1/12,000 years. The deeply uncertain boundary conditions are (i) the future average sea level rise and (ii) the storm surge intensity. The probability of dike failure conditional to the deeply uncertain boundary conditions is calculated using a semi-probabilistic approach to dike failure mechanism, by use of the van-der-Meer model [50], where the hydraulic boundary conditions at extremes are identified using the "de Haan" method [53].

We consider failure of dike body only. The dike body is modeled as a single dike section. Of the different mechanisms of dike body failure, we consider wave run-up overtopping only. This process is the dominant mechanism of body dike failure for Afsluitdijk [54]. Dike failure occurs when the occurring overtopping discharge is larger than the critical overtopping discharge. Equation (A1) defines the probability of dike failure due to overtopping.

$$P(F) = P(m_o \cdot q_{occ} > m_c \cdot q_{crit}) \tag{A1}$$

In Equation (A1), F is the event "failure of the dike", $q_{occ}$ is the occurring overtopping discharge, $q_{crit}$ is the critical overtopping discharge, and $m_o$ and $m_c$ are stochastic variables that represent the uncertainty in the process.

The critical overtopping discharge is a given project parameter [54]. The occurring overtopping discharge is calculated as in Equation (A2).

$$q_{occ} = \min(q_b, q_n) \cdot \sqrt{g \cdot h_s{}^3} \tag{A2}$$

In Equation (A2) $q_b$ is discharge due to breaking waves, $q_n$ is the discharge due to non-breaking waves, and $h_s$ is the characteristic wave height.

Discharge of breaking and non-breaking waves is calculated as in Equation (A3).

$$q_b = 0.067 / \sqrt{\tan(\alpha) \cdot b \cdot \xi \cdot \exp\left(\frac{C_4 \cdot RC}{h_s \cdot \xi \cdot b \cdot f \cdot \beta}\right)} \tag{A3a}$$

$$q_n = 0.2 \cdot \exp\left(\frac{C_5 \cdot RC}{h_s \cdot f \cdot \beta}\right) \tag{A3b}$$

In Equation (A3), $h_s$ are hydraulic boundary conditions, $RC$ and $\xi$ are model parameters, $\alpha$, $\beta$ and $f$ are constants, and $C_4$ and $C_5$ are stochastic variables that represent the uncertainty in the process.

Model parameters of Equation (A3) are calculated as in Equation (A4).

$$RC = CH - (SWL_{2050} + SS) \tag{A4a}$$

$$\xi = \tan(\alpha)/\sqrt{s_0} \tag{A4b}$$

$$s_0 = \frac{2 \cdot \pi \cdot h_s}{g \cdot T_s \cdot 2} \tag{A4c}$$

In Equation (A4), $RC$ is the crest free board, $\xi$ is the relative wave run-up, $s_0$ is the wave steepness. Inputs of Equation (A4) are defined in Equation (A5). Equation (A5) define the hydraulic boundary conditions.

$$SS \sim GEV(\mu_{SS}, \sigma_{SS}) \tag{A5a}$$

$$SWL_{2050} = SWL_{2017} + SLR \cdot (2050 - 2017) \tag{A5b}$$

$$h_s = C_1 \cdot \log(SS) + C_2 \tag{A5c}$$

$$T_m = \sqrt{h_s/C_3} \tag{A5d}$$

In Equation (A5), $SS$ is the storm surge intensity, sampled from a GEV Type I (i.e., Gumbel) distribution with location and scale parameters $\mu_{SS}$ $\sigma_S S$. The location parameter is considered deeply uncertain, hence different values are tested. The scale parameter is estimated from historical data. $SWL_{2017}$ is the present average sea water level (2017), and $SLR$ is the average sea level rise rate. The $SLR$ parameter is considered deeply uncertain, hence different values are tested. $h_s$ and $T_m$ are the height of wave characteristic and the spectral wave period. Parameters $C_1$, $C_2$ and $C_3$ are identified at local conditions according to the "de Haan" method, considering the asymptotic dependency at extremes [53].

The model is simulated using an "importance sampling" Monte Carlo method. In the importance sampling approach, we simulate the system for storm surge events, $SS_{IS}$, with a return period larger than the base year, set to 2000 year, such that $SS_{IS} = SS > \widehat{SS}$, where $\widehat{SS} = GEV^{-1}(1 - 1/\text{base year})$. The actual frequency of failure (in yrs$^{-1}$) is estimated from the frequency of failure calculated in the space of importance sampling (unitless) by dividing the latter by the base year. The sample size for each simulation run is 10,000.

**Appendix B. Signposts Observational Uncertainty**

All signposts are defined as average over a moving window, therefore they converge to normal distribution. We assume a quasi-stationary condition, i.e., the rate of change is negligible, therefore signposts can be considered as unbiased estimator of the parameters that they track.

The distribution of signpost $S_A$ is identified in Equation (A6).

$$S_A \sim \mathcal{N}\left(SLR, \frac{\sigma_{SWL}^2}{VAR([1..10])}\right) \tag{A6}$$

where $SLR$ is the sea level rise rate, $\sigma_{SLR}$ is the observational uncertainty of the yearly value, estimated at den Oever station, and $[1..10]$ the integer interval between 1 and the length of the average moving window.

The distribution of Signposts $S_E$ and $S_{RE}$ are identified from their empirical distribution. The empirical distributions are identified from the definition of these signposts, sampling from $SS$. Equations (A7) and (A8) are the definitions of $S_E$ and $S_{RE}$.

$$S_E \quad = \quad \frac{1}{w} \sum_{y}^{w} MWL_y \tag{A7}$$

$$S_{RE} \quad = \quad \frac{1}{w \cdot n} \sum_{i}^{n} \sum_{y}^{w} MWL_{y,i} \tag{A8}$$

In Equations (A7) and (A8), $MWL_y$ is the maximum water level at year $y$, $w$ is the length of the moving window (i.e., 20 years), $n$ is the number of stations (i.e., 5), $MWL_{y,i}$ is the corrected maximum yearly water level, and $y$ and $i$ are the index of past year and station. $MWL_y$ is estimated as in Equation (A9).

$$MWL_y(SLR, SS) = SWL_{2050}(SLR) + SS_y \tag{A9}$$

In Equation (A9), the yearly max water level is the consequence of the superposition of the average sea water level at that year and the storm surge process, considered independent. The average sea water level is estimated as in Equation (A5b), hence it grows with the sea level rise rate. The observational uncertainty of average sea level is considered to be negligible with respect to the variance of the maximum yearly storm surge, hence the variance of $S_E$ and $S_E$ depends only on the latter and on the number of yearly data-points.

**Appendix C. Estimation of Consequences**

With respect to the adaptation required for the Path Protection, the consequence of regretful adaptation is the cost of enriching the dike, estimated in [51] at 380 million euro. The consequence of missed adaptation is the total additional risk of being at the adaptation tipping point. Risk is defined as the probability of dike failure multiplied by its consequences; additional is defined with respect to the situation at the current conditions. [51] estimates the total risk at current condition due to failure of Afsluitdijk at 320 million euro. For Afsluitdijk, the probability of failure at the adaptation tipping point is twice the probability of failure at current conditions, therefore the risk is double. Consequently, the total additional risk of being at the adaptation tipping point is the same amount of the total risk at current condition.

**Appendix D. Software and Data Availability**

The code used in the experiment can be freely downloaded at GitHub page of the corresponding author, i.e., https://github.com/luciofaso/Monitoring_DAP, released under the MIT license. The data used in the experiment can be freely downloaded at the data portal of Rijkswaterstaat, i.e., https://waterinfo.rws.nl.

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
