# Peer review of "Assessing the Capacity of Adaptive Policy Pathways to Adapt on Time by Mapping Trigger Values to Their Outcomes"

_sustainability, doi:10.3390/su11061716_

Round 1
Reviewer 1 Report
Thank you for the opportunity to review this article. I believe that the scientific concepts and research presented here are original and well defined and represents a significant and worthwhile advance. Additionally, the article is well organized and interesting. Overall I think the article is an excellent candidate for publication.
That said, I have one scientific concern and a number of minor issues that should probably be resolved.
I would like to see more discussion about uncertainty overall. Particularly, the uncertainty associated with the Se trigger (based on a 20-year moving average) and the related peak sea levels resulting from the Gumbel distribution discussed in the appendix. The article notes that the safety standard for the dike is 1 in 5,000 or 1 in 12,000. It is not clear how 10,000 samples, even if using importance sampling montecarlo methods, is adequate. The reference to return periods greater than 2,000 years is also confusing. A discussion of the relationship to changes in the Gumbel distribution parameters to the observed metric (a 20 year moving average of highest surge levels) would be helpful as well. The presentation of figures 5-9 and associated discussion presents compelling arguments for the value of the proposed mappings, but the visual display of sharp lines without confidence limits implies a high degree of confidence in the results.
Additionally, the focus on probability (vertical axis in the figures) of dike failure versus economic investment (for dike modification) and damage potentially obscures the dollar impact of the relationship between regret and missed adaptation. I would think that damage associated with a worst-case failed dike is probably orders of magnitude higher than the proposed levels of dike investment, and the potential for a skewed result based on any potential non-linearity in the relation between probability and damage.
Lastly, some discussion about the sensitivity of the result to the GEV equations assumed might be important. What happens if we assume a GEV distribution with a fat tail? I would be interested in the author's comment on this, whether or not included in the article.
I found many language use issues that I would like to see improved prior to publication. I think the article would benefit from a very careful edit for grammar and style overall.
Many instances of passive voice, as in:
The abstract, lines 37, 39, 87,
241 (where it becomes difficult to distinguish that monitoring of storm intensity is something that the authors are proposing or something that is already in the works.)
There are some tense changes in the article. For instance:
209 “In the following, we will present…”
224,
273 (where tense probably should be future)
Some odd usage and grammar errors, as in:
58 “In the literature, it has been suggested to base the monitoring system” is awkward.
200 “…possibly by improving the monitoring system, otherwise by redefining…” might be clearer as “…possibly by improving the monitoring system, or by redefining…”
203 Commas before and after for example, as in, for example, …
215 “…the most inhabited and economic valuable areas…” should probably be “…the most inhabited and economically valuable areas…”
164, “When the consequences of adaptation decisions are available as in Table 1, one can identify from these the trigger-consequences mapping.” Is odd oddly phrased
233 comma usage “…can increase or, if global policies…” should be a comma after increase. . Also, will the rate of sea-level increase actually decline from current rates in any climate scenarios? Perhaps there should be a citation here.
The double-ended arrows in figure 2 used to highlight the potential value of adaptation might be improved by highlighting in the figure that the top of the arrow corresponds to the maximum of regret ($2M) perhaps through the use of a dotted line?
Author Response
Please see attached file.
LR

Reviewer 2 Report
Assessing the capacity of adaptive policy pathways to adapt on time by mapping trigger values to their outcomes, Raso et al. Sustainability
This paper develops a statistical approach to selecting adaptation triggers based on the tradeoff between Type I and Type II errors - regretful adaptation and missed adaptation, respectively. The method is demonstrated for a sea level rise case study in the Netherlands.
The topic is extremely important, interesting, and well within scope for the journal. The concept of setting adaptation thresholds is relevant for a broad set of planning problems. As this is a statistics-focused paper, my concerns are related to uncertainty estimates. I will recommend major revisions, but some of these points may require only clarification and framing rather than changes to the experiment.
1) The method assumes that the probabilities of future scenarios are known. This is not in itself a problem, and I do not believe a purely "deep uncertainty" framing should be required. However, assuming perfect knowledge of a PDF does require at least some discussion. Do we estimate these distributions from a GCM ensemble? Is it possible that we could account for uncertainty in the PDF itself? The appendix describes how the probabilities are estimated empirically for this experiment, which is a crucial aspect of the paper because it defines how the method should be used. There may be room to include this in Section 2 instead.
One idea might be an experiment to test the sensitivity of the minimum-cost trigger value to the pdf assumptions. For example, if the mean and/or variance of the underlying distribution were changed by +/- 10%, how would the trigger value change?
2) Even in the absence of climate change, under a stationary flood risk distribution, there would be tremendous sampling uncertainty in this analysis. The authors define Type I/II errors related to the 5000 and 10000-year flood events using what appears to be only a 20-year moving window. Yet these errors and their consequences do not account for any sampling uncertainty in these estimates that would arise due to the limited record. Would this change the analysis, and if so, how? Experience suggests this may be a more severe effect than the "deep uncertainty" in the pdf estimate.
3) In general, the analysis recalls the Van Dantzig 1956 levee problem, especially the trigger-consequence mapping where costs are minimized. The only difference is that here the authors consider a nonstationary world and are looking for trigger points rather than an optimized levee height. There was recently a paper by Oddo et al. 2017 who showed the range of expected cost curves that could occur under different uncertainty assumptions, using a variant of the same Van Dantzig problem. I wonder whether this paper could have a similar component, where the trigger-consequence mappings account for uncertainty in the pdfs and also sampling uncertainty, aligned with the comments above.
Van Dantzig D. Economic decision problems for flood prevention. Econometrica, 1956; 24(3):276–287.
Oddo, Perry C., et al. "Deep Uncertainties in Sea‐Level Rise and Storm Surge Projections: Implications for Coastal Flood Risk Management." Risk Analysis (2017).
4) Another experimental aspect that is unclear from the paper, but could be very important for the real-world problem of climate adaptation, is the use of information at different lead times. The choice of lead time would define the tradeoff between a more certain estimate of the PDF and having time to adapt with infrastructure measures. It is at least worth a discussion here.
5) A minor point, it is odd that the title does not include anything about probability or statistics, considering the methodological contributions of the paper are almost purely statistical in nature. The concept of mapping trigger values to their outcomes is only a simulation question, and it does not describe the contribution very well.
Author Response
Please see attached file.
LR

Reviewer 3 Report
The paper is on a very interesting topic, and is worth publication with major revisions. The authors propose a new methodology for risk-informed decision making which evaluates the effectiveness of given metric (referred to as signpost) for on-time (neither early nor late) policy adaptation. The overall presentation can be improved. The article does not read very smooth; it is sometime confusing, and often long unnecessary statements are used. Here are some general comments to improve the quality of the work. The proposed method claims to offer criteria for developing timely policy. There are two main critics for the approach: 1) The success of this approach is closely related to the precision of two probability curves, i.e., regretful action and missed adaption. The uncertainty associated with the estimation of these curves, which is key to success of proposed methodology is not discussed. 2) The system only proposes the start of an “adaptive and timely policy” when certain thresholds reaches, and has no criteria for judging whether a policy can be adapted timely. How would the system can say it is timely? For example, the future sea level rise can go in a second or in a century under various scenarios, how is this uncertainty incorporated in the system, if it is at all. The methodology is very abstract, and could be better explained. Terms such as trigger, signpost, critical uncertainties, tipping points can be explained with examples where first introduced in the methodology. It is especially important to give readers a sense of what is what, and how they are related. Some non related sentences can be moved to discussion section, and some essential equations can be presented in methodology instead of appendix, especially those explaining how a signpost and critical uncertainty (better to be called variables) are related. The application section can be renamed to results and discussion, the discussion is indeed a major shortcoming of the paper that is missing. Dissuasion on under what condition the system may work best and under what conditions it may not with some concrete examples. Discussion on multiple triggers, briefly discussed in the methodology section can be presented better in result section with improved explanations and concrete examples. The use of trigger and signpoint is a bit confusing in presentation. Triggers are indeed the threshold values for signpoints. In the manuscript, while explaining trigger-probability and trigger-consequence graphs, triggers are sometimes referred to as “possible triggers”, and other times without the term “possible”. This is especially confusing as the horizontal axis in figures represents “signpoint” or “possible trigger”, not “trigger” alone. The use of signpoint or possible trigger is recommended. In current presentation, figures are sometimes not presented in the same page as they first mentioned, which makes reading rather difficult. Figures 4, 5, 8, 7 can all be presented as one figure with a b c d. It makes the comparison much easier. The captions and symbols need to be self explanatory, and readers should not go back to the manuscript to find what is what. Table 2 can also be presented as a legend in the merged new figure. Other miner comments: Please double check the use of “a” and “an” in the manuscript. Please double check the use of third person verb (“s”) in present tense, and some past tense verbs. L 65: Simplify and clarify the sentence: Despite these initial research efforts, the evaluation of the capacity of monitoring system to enable timely adaption suffers from a lack of methods and instruments fitted for this purpose. L91: Simplify and clarify the sentence: The analyst that wants to ensure this condition has to test whether monitoring system will be able to provide a timely adaptation signal. Fig 2: SE and trigger values in (m) were introduced yet. L161: Multiple tipping points: what are the examples, how it is even possible, it is more like discussion than methodology. L228: Critical uncertainty is indeed “variables”, a term which is later used in the manuscript to describe example of critical uncertainty.
Author Response
Please see attached file.
LR

Round 2
Reviewer 2 Report
The authors have responded to all comments from the previous round. A number of the concerns that I had were clarified by the authors. I recommend the paper to be accepted for publication.